# Comparison of Long-Run Net Returns of Conventional and Organic Crop Rotations

**Michael R. Langemeier** [1,*] **, Xiaoyi Fang** [1] **and Michael O'Donnell** [2]

1   Department of Agricultural Economics, Purdue University, West Lafayette, IN 47907, USA; fang232@purdue.edu
2   Purdue Extension, Purdue University, West Lafayette, IN 47907, USA; modonnel@purdue.edu
*   Correspondence: mlangeme@purdue.edu

**Abstract:** This study compares the long-run net returns to land of conventional corn/soybean and corn/soybean/wheat crop rotations to that of an organic corn/soybean/wheat crop rotation. The net returns to land for the organic crop rotation were found to be approximately $68 and $74 per acre higher than those of the conventional corn/soybean and conventional corn/soybean/wheat crop rotations, respectively. Average net return estimates are sensitive to price, yield, and cost assumptions. Organic crop prices would have to drop more than 17.8 percent and organic crop yields would have to drop more than 16.8 percent before the conventional corn/soybean crop rotation was more profitable than the organic corn/soybean/wheat crop rotation. These percentage changes are relatively small compared to the historical relationships between organic and conventional crop prices and yields. A risk model was used to examine the trade-off between expected net returns and downside risk. Converting even a small proportion of acreage to an organic corn/soybean/wheat crop rotation improves net returns and reduces downside risk compared to only utilizing conventional crop rotations.

**Keywords:** organic crops; net returns; risk

## 1. Introduction

Due to continued increases in demand for certified organic grains, crop farmers that have transitioned from conventional grains to certified organic grains report higher net returns per acre [1–3]. Despite this, certified organic land accounts for less than two percent of U.S. farmland [4]. There are several factors for potential organic farms to consider when they are examining the transition from conventional to organic production. Peterson et al. [5] indicated that this transition may be related to profitability, environmental stewardship, an organic lifestyle, or a combination of these three motivations. Crop producers that are considering transferring acreage to organic production need to examine each of these factors. The variability of net returns to farm operations that results from fluctuating yields, costs, and prices is also an important factor when examining organic crop production.

Even though there have been a few studies that have examined the relative profitability prospects for organic crop production compared to conventional crop production, in general, information pertaining to relative profitability is lacking. McBride et al. [1] examined the profitability of conventional and organic corn, soybean, and wheat production using national survey data. Based on their analysis of the surveys, they found that significant economic net returns were possible from organic production of corn, soybeans, and wheat. The primary reason for the higher net returns were price premiums for organic crops. Economic costs and yields were lower for organic crops. The transition to organic crops was challenging due to such factors as achieving effective weed control and processes involved with organic certification.

Crop price, yield, cost, and net return data are available from the University of Minnesota FINBIN database [6]. Using FINBIN data from 2014 to 2018, organic crops tend to have lower yields, receive a higher price, and have higher costs per acre.

The studies above did not include information pertaining to the net returns during the transition phase. During the transition phase, a farm utilizes organic crop practices but typically only receives conventional crop prices. The transition period will impact potential net returns and the variability of net returns of the organic crop rotation. Because of this, comparisons of long-run net returns of conventional and organic crop rotations should include the transition phase in the analysis of the organic crop rotations.

The objective of this study is to compare the long-run net returns to land of conventional corn/soybean and corn/soybean/wheat crop rotations to that of an organic corn/soybean/wheat crop rotation. The organic crop rotation examined in this study included a three-year transition period.

## 2. Materials and Methods

### 2.1. Enterprise Budgets

Ten-year enterprise budgets for each crop enterprise were developed so that we could compute net returns to land for conventional and organic crop rotations. Land was assumed to represent average productivity cropland for Indiana. Enterprise budgets included conventional corn, conventional soybeans, conventional wheat, transition soybeans, transition wheat, organic corn, organic soybeans, and organic wheat. Each enterprise was developed using projected prices, yields, government payments, and costs. Conventional corn, soybean, and wheat enterprises were used to estimate net returns per acre for a corn/soybean and a corn/soybean/wheat crop rotation. Transition soybeans and transition wheat were used along with conventional corn, organic corn, organic soybeans, and organic wheat to estimate net returns per acre for an organic corn/soybean/wheat crop rotation. The inclusion of conventional corn, transition soybeans, and transition wheat in the organic crop rotation reflects the fact that a transition period is needed before a farm can produce organic crops. The transition was assumed to take place over time rather than just the first three years of the ten-year period. Specifically, six fields were targeted for organic crop production. One-third of the fields were transitioned per year, resulting in a complete transition to organic crop production in year 5. The timing of the transition of the fields was set up so that the third crop produced was eligible for organic certification. Because corn has historically been the most profitable organic crop, the first organic crop within the organic crop rotation was corn.

Table 1 lists the crop prices that were used for year 1 and years 2 through 10. After being lower in the first year (i.e., 2020), crop prices were assumed to stabilize and reach a long-run equilibrium. The historical difference between conventional and organic prices, generated using FINBIN data from 2014 to 2018, was used to estimate organic prices.

**Table 1.** Crop Price Assumptions for Conventional, Transition, and Organic Enterprise Budgets.

| Crop | Year 1 | Years 2–10 |
|---|---|---|
| Conventional Corn | $3.00 | $3.60 |
| Conventional Soybeans | $8.00 | $8.85 |
| Conventional Wheat | $4.90 | $4.85 |
| Transition Soybeans | $8.00 | $8.85 |
| Transition Wheat | $4.90 | $4.85 |
| Organic Corn | $6.90 | $8.30 |
| Organic Soybeans | $16.80 | $18.60 |
| Organic Wheat | $9.25 | $9.70 |

FINBIN data from 2014 to 2018 were also used to estimate organic crop yields. Crop yields for the first year and the annual increment in yields (i.e., trend yield adjustments) used for the

remaining budget years are reported in Table 2. Yield drags for the transition and organic crops were approximately 33 percent.

**Table 2.** Crop Yield Assumptions for Conventional, Transition, and Organic Enterprise Budgets.

| Crop | Year 1 (bu/acre) | Annual Increment (bu/acre) |
|---|---|---|
| Conventional Corn | 176.0 | 2.00 |
| Conventional Soybeans | 54.0 | 0.50 |
| Conventional Wheat | 77.0 | 0.35 |
| Transition Soybeans | 36.3 | 0.35 |
| Transition Wheat | 58.5 | 0.25 |
| Organic Corn | 118.8 | 1.50 |
| Organic Soybeans | 36.3 | 0.35 |
| Organic Wheat | 58.5 | 0.25 |

In addition to crop revenue, computed by multiplying crop price by crop yield, the gross revenue for each enterprise included government payments and crop insurance indemnity payments. Most government payments are based on base acres rather than crop acres, so farms that transition to organic production will continue to receive government payments.

Cost estimates for each crop were derived using FINBIN [6], Chase et al. [7], Klein et al. [8], and Langemeier et al. [9]. Costs were broken down into two major categories: variable costs and fixed costs. Variable costs included seed, fertilizer, manure, herbicide, insecticide, crop insurance, general farm insurance, repairs, fuel, and operating interest. Fixed costs included labor costs and depreciation and interest on machinery and equipment. The first-year budget includes detailed cost estimates. Budgets for years two through ten were created using percentage adjustments to the first-year budget estimates. The organic crop budgets recognized the substitution of manure, extra tillage operations, and additional labor for fertilizer, herbicide, and insecticide. In turn, variable costs were generally lower and fixed costs were generally higher for the organic crops. To reflect what needs to take place to attain organic certification, the transition budgets use the cost estimates of the organic crops. More detail pertaining to the layout of the enterprise budgets can be found on the Center for Commercial Agriculture website [10].

Average gross revenue, variable cost, contribution margin, fixed costs, earnings, and net return to land were computed for each crop rotation. The contribution margin was computed by subtracting variable costs from gross revenue, which includes crop revenue, government payments, and crop insurance indemnity payments. Earnings were computed by subtracting variable and fixed costs from gross revenue. To account for differences in the timing of revenue, costs, and net returns between the crop rotations, gross revenue, variable cost, contribution margin, fixed costs, earnings, and net return to land during the ten-year period were discounted using net present value analysis [11]. A discount rate of 6 percent was utilized for the net present value computations. With respect to the timing of net returns, the net returns for the organic crop rotation during the last five years of the ten-year period, which took place after the transition, were much larger than the net returns during the first five years, which included the transition period.

Annual discounted net returns to land were used to examine break-even prices and yields for the organic crop rotation and to utilize the downside risk model described below. The break-even prices and yields were computed by comparing the net present value of annual net returns to land between the conventional and organic crop rotations. Specifically, each break-even price or yield represents a scenario in which the annual discounted net returns to land are equivalent between the two crop rotations being examined.

*2.2. Downside Risk Model*

Variability and downside risk are commonly used to measure risk in production agriculture [12]. Variability focuses on dispersions from the mean, while downside risk focuses on low outcomes.

To account for potential low net returns during the transition years, downside risk was the focus in this study. Expected net returns to land and risk for the three crop rotations, as well as for combinations of the crop rotations, were examined using a downside risk model. The Target MOTAD model maximizes expected net return subject to a constraint or limit on the total negative deviations measured from a fixed target or target income [13,14]. The Target MOTAD model focuses on the downside risk that occurs when the net return for an individual year falls below a target level or net return. As with other portfolio models, trade-offs between risk, as measured by total negative deviations below a target net return, and expected net returns are examined. To compute total negative deviations below the target net return, net returns for each crop rotation in each year were compared to the target net return. A negative deviation occurred when net returns for a given year were lower than the target net return. These annual deviations were then summed for each crop rotation to obtain the total negative deviations below the target net return for a crop rotation or combination of crop rotations.

The solution of the Target MOTAD model that identifies the maximum expected net return also has the highest level of total negative deviations below the target net return. In other words, this is the profit-maximizing solution. As the total negative deviations below the target net return become more constrained, risk and expected net return decline. A target net return to land of $115 per acre, which is the average annual net present value of net return to land for the conventional corn/soybean crop rotation, was used for the analysis in this study. The net present value of annual net return per acre for each crop rotation was used to generate the risk and return trade-off.

## 3. Results

### 3.1. Net Returns per Acre and Break-even Prices and Yields

Average discounted gross revenue, contribution margin, earnings, and net returns to land per acre for the conventional and organic crop rotations are presented in Table 3. The gross revenue for the organic crop rotation was significantly higher than the gross revenue for the conventional corn/soybean and corn/soybean/wheat crop rotations. Variable cost per acre was relatively lower for the organic crop rotation, but fixed costs were relatively higher. Essentially, the organic crop rotation substitutes manure and machinery costs for fertilizer, herbicide, and insecticide costs. Labor costs were higher for the organic crop rotation. The net return to land for the organic crop rotation was $183 per acre, or approximately $68 and $74 higher than that of the conventional corn/soybean and corn/soybean/wheat crop rotations, respectively.

**Table 3.** Average Net Returns per Acre for Conventional and Organic Crop Rotations.

| Enterprise | Conv | | Org |
| --- | --- | --- | --- |
| | C/S | C/S/W | C/S/W |
| Gross Revenue | 442.86 | 392.16 | 505.56 |
| Variable Cost | 253.88 | 212.02 | 196.35 |
| Contribution Margin | 188.97 | 180.14 | 309.21 |
| Fixed Cost | 233.25 | 230.79 | 285.71 |
| Earnings | −44.28 | −50.65 | 23.50 |
| Net Return to Land | 115.13 | 108.75 | 182.90 |

Definitions: C = Corn, S = Soybeans, W = Wheat.

The average net returns to land reported in Table 3 are sensitive to changes in relative crop prices, yields, and costs. Break-even prices and yields were computed by comparing the net present value of net returns to land among the three crop rotations. The results are presented in Table 4.

**Table 4.** Breakeven Organic Crop Prices and Yields [1,2].

| Crop. | Base Case | C/S NR | C/S/W NR |
|---|---|---|---|
| Organic Crop Prices, Years 2–10 (per bushel) | | | |
| Corn | $8.30 | $6.82 | $6.68 |
| Soybeans | $18.60 | $15.29 | $14.97 |
| Wheat | $9.70 | $7.97 | $7.81 |
| Transition and Organic Crop Yields, Year 1 (bushels per acre) | | | |
| Corn | 118.8 | 98.8 | 96.9 |
| Soybeans | 36.3 | 30.2 | 29.6 |
| Wheat | 58.5 | 48.7 | 47.7 |

[1] The C/S NR column represents the price and yield needed to equilibrate net returns for conventional corn/soybean and organic crop rotations. [2] The C/S/W NR column represents the price and yield needed to equilibrate net returns for conventional corn/soybean/wheat and organic crop rotations.

The first part of Table 4 shows the organic crop prices used in Table 1 for years 2 through 10 (referred to as the base case), and what the organic crop prices would need to be for the net returns to land for the organic crop rotation to equal those of the two conventional crop rotations. Holding crop yields and costs constant, organic crop prices would need to be reduced 17.8 percent and 19.5 percent for the average net return to land to equal that of the average net returns for the conventional corn/soybean and conventional corn/soybean/wheat crop rotations, respectively.

The second part of Table 4 presents the transition and organic crop yields used for the base case as well as the break-even crop yields. The table only shows yields for the first year. Trend yield adjustments are made to estimate crop yields in years 2 through 10, so any changes in the projections for the first year will change crop yields for the entire ten-year period. Holding crop prices and costs constant, transition and organic crop yields would need to decrease 16.8 percent and 18.4 percent for the average net return to land of the organic crop rotation to equal those of the conventional corn/soybean and conventional corn/soybean/wheat crop rotations, respectively.

### 3.2. Trade-Off between Risk and Return

Using the Target MOTAD model, the trade-offs between risk, as measured by the total negative deviations below the target net return of $115 per acre, and expected net return are listed in Table 5. In addition to tracing out the frontier (i.e., relationship between risk and return), this table illustrates the expected net return and risk for each crop rotation, and for a scenario (L Org = 0.100) in which the organic crop rotation is limited to 10 percent of the farm's acreage.

**Table 5.** Expected Net Return to Land and Total Negative Deviations below Target Income ($ per Acre).

| Scenario | Expected Net Return | Negative Deviations | Conventional C/S | Conventional C/S/W | Organic C/S/W |
|---|---|---|---|---|---|
| 1 | 182.90 | 133.73 | 0.000 | 0.000 | 1.000 |
| 2 | 178.51 | 125.00 | 0.065 | 0.000 | 0.935 |
| 3 | 173.49 | 115.00 | 0.139 | 0.000 | 0.861 |
| 4 | 168.46 | 105.00 | 0.213 | 0.000 | 0.787 |
| 5 | 163.43 | 95.00 | 0.287 | 0.000 | 0.713 |
| 6 | 158.40 | 85.00 | 0.362 | 0.000 | 0.638 |
| 7 | 153.37 | 75.00 | 0.436 | 0.000 | 0.564 |
| 8 | 148.35 | 65.00 | 0.510 | 0.000 | 0.490 |
| 9 | 143.32 | 55.00 | 0.584 | 0.000 | 0.416 |
| 10 | 138.29 | 45.00 | 0.658 | 0.000 | 0.342 |
| 11 | 133.19 | 35.00 | 0.700 | 0.031 | 0.269 |
| 12 | 126.92 | 25.00 | 0.224 | 0.550 | 0.226 |
| C/S = 1 | 115.13 | 72.98 | 1.000 | 0.000 | 0.000 |
| C/S/W = 1 | 108.75 | 99.75 | 0.000 | 1.000 | 0.000 |
| Org = 1 | 182.90 | 133.73 | 0.000 | 0.000 | 1.000 |
| L Org | 121.91 | 41.37 | 0.900 | 0.000 | 0.100 |

Before discussing the risk and return trade-off results illustrated in scenarios 1 through 12, we will discuss the expected net return and risk for each crop rotation. The organic crop rotation had the highest expected net return, but also exhibited more risk. The downside risk for the organic crop rotation was exasperated by the low net returns obtained during the transition period. The conventional corn/soybean crop rotation had a higher expected net return and less downside risk than the corn/soybean/wheat crop rotation. Unless some combination of the corn/soybean/wheat crop rotation with the other two crop rotations reduces risk, we would not expect to find this crop rotation utilized for scenarios pertaining to the risk and return trade-off. The net return to land in the first year of the analysis was higher for the corn/soybean/wheat crop rotation than it was for the corn/soybean crop rotation, so the corn/soybean/wheat crop rotation could be utilized to reduce downside risk during this year.

The risk and return trade-off is represented by scenarios 1 through 12 in Table 5. The scenario that maximized expected net returns (i.e., scenario 1) had the highest risk level and utilized the organic corn/soybean/wheat (Organic C/S/W) crop rotation. To trace out the remainder of the frontier, the risk level was reduced in increments of 10, reaching the lowest risk level in scenario 12. As risk reduced from scenarios 2 through 10, the proportion of acreage in the conventional corn/soybean (Conventional C/S) crop rotation increased while the proportion of acreage in the Organic C/S/W crop rotation decreased. To reduce risk below a risk level of 45, it was necessary to incorporate the conventional corn/soybean/wheat (Conventional C/S/W) crop rotation into the acreage mix. In scenario 11, a risk level of 35, 3.1 percent of the acreage was devoted to Conventional C/S/W production. For the lowest risk level (i.e., scenario 12), 55.0 percent of the acreage involved Conventional C/S/W production, 22.6 percent involved Organic C/S/W production, and the remaining 22.4 percent of the acreage was devoted to Conventional C/S production. The results in Table 5 indicate that it is necessary to reduce Organic C/S/W acreage to mitigate downside risk.

Most farms that are transitioning to an organic crop rotation will only transition a portion of their acreage so that they can learn the system and reduce risk. To account for this fact, we illustrate a scenario (L Org) in Table 5 that allows for only 10 percent of a farm's acreage to transition to organic crop production. As you would expect, the expected net return for this scenario was lower and the risk was higher than in scenarios 11 and 12, in which the organic crop rotation was not constrained. The acreage constraint reduced expected net return and increased risk. However, the expected net return for this scenario was higher than the expected net return for the corn/soybean crop rotation, and the risk for this scenario was much lower than the risk exhibited by any of the individual crop rotations. Thus, converting even a modest proportion of a farm's acreage to an organic crop rotation dramatically reduces risk and at the same time increases expected net returns.

## 4. Discussion

This study compared long-run net returns to land of conventional corn/soybean and corn/soybean/wheat crop rotations to that of an organic corn/soybean/wheat crop rotation. The net returns to land for the organic crop rotation were found to be approximately $68 and $74 higher than those of the conventional corn/soybean and conventional corn/soybean/wheat crop rotations, respectively. Break-even organic crop prices and yields were computed by comparing the net present value of net returns to land among the three crop rotations. Crop prices and yields would need to fall 17 to 18 percent for the average net return to land of the organic crop rotation to equal that of the conventional corn/soybean crop rotation. A risk model was used to examine the trade-off between expected net returns and downside risk. Transitioning a portion of acreage to organic crop production was shown to dramatically reduce risk and increase expected net returns.

The enterprise budgets described in this study could be used by farms considering transitioning a portion of their acreage to certified organic crop production. Producers considering this transition should carefully examine the sensitivity of net returns to alternative price, yield, and cost assumptions. It is also important to recognize that the crops grown, manure used, and tillage practices vary



substantially among organic crop farms. Furthermore, anything a farm could do to mitigate the low net returns, which increases downside risk, during the two transition years would be helpful when adopting an organic crop rotation. This may include, but is not necessarily limited to, examining alternative crops or alternative markets for the crops utilized during the transition. Alternative crops may involve forage-based rotations and cover crops that could be grazed. Alternative markets may involve producing non-GMO crops, which typically result in higher crop prices.

In general, farms have considerably more experience with producing conventional crops than they do producing organic crops. This may create challenges when transitioning a portion of their acreage to organic production. FINBIN data (Center for Farm Financial Management, 2020) show a much wider difference in enterprise net returns among organic crop farms and their conventional counterparts. This wider difference is likely due to the difficulty of managing an organic crop system and the learning curve associated with growing organic crops.

**Author Contributions:** M.R.L., X.F., and M.O. conducted the conceptualization, methodology, investigation, writing, and editing of this article jointly. All authors have read and agreed to the published version of the manuscript.

**Funding:** This research received no external funding.

**Acknowledgments:** The authors would like to acknowledge support from the Center for Commercial Agriculture in the Department of Agricultural Economics at Purdue University for the work in this article.

**Conflicts of Interest:** The authors declare no conflict of interest.

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
