# Peer review of "Comparison of Long-Run Net Returns of Conventional and Organic Crop Rotations"

_sustainability, doi:10.3390/su12197891_

Round 1

Reviewer 1 Report

The manuscript titled "Comparison of Long-Run Net Returns of Conventional and Organic Crop Rotations" is very well written. The results are presented in a clear and good manner. However, the importance of study is not clearly mentioned in the abstract. I would recommend to add 1-2 sentences explaining the importance of this study in the abstract section. 

I liked the idea of providing recommendations on how to mitigate low net returns during the transition years (L210-212) in the manuscript. I would suggest to elaborate on these recommendations.

Author Response

The manuscript entitled “Comparison of Long-Run Net Returns of Conventional and Organic Crop Rotations” is very well written. The results are presented in a clear manner. However, the importance of the study is not clearly mentioned in the abstract. I would recommend adding 1 to 2 sentences explaining the importance of this study in the abstract section.

Per your suggestion, we added a sentence pertaining to our results to the abstract. We also reworded the final sentence in the abstract. In addition, we elaborated on the results with respect to converting 10 percent of a farm’s acreage to an organic rotation on lines 221-226.

I liked the idea of providing recommendations on how to mitigate low net returns during the transition years (lines 210-212) in the manuscript. I suggest that you elaborate on these recommendations.

We elaborated on the mitigation of low returns during the transition period in the discussion section. This text can be found on lines 243-248.

Additional Reply:

In the process of addressing comments from another reviewer would have added text to the methods, results, and discussion sections. These additions, along with addressing your comments, have improved the manuscript.

Reviewer 2 Report

Major Comments

The paper aims to address the variation in the long-run returns to land of conventional and organic crop rotation. Though not original, the paper aimed to compare the long-run net returns to land of conventional and organic corn/soybean/wheat crop.

The study is relevant to the Journal of Sustainability. However, the methodology is not sufficient for the analysis making the results doubtful. The conclusions are also not clearly presented.

Minor Comments

Line 4-10: correct the 1-3 indication as 1 and 2.

Line 218 -219: A clear conclusion and extra information will provide more clarity on the method and analysis carried out.

Author Response

This paper aims to address the variation in the long-run returns to land of conventional and organic crop rotations. Though not original, the paper aimed to compare the long-run net returns to land of conventional and organic corn/soybean/wheat crop rotations.

The study is relevant to Sustainability. However, the methodology is not sufficient for the analysis making the results doubtful. The conclusions are also not clearly presented.

In response to your concerns, we have rewritten the materials and methods, and discussion sections. Specifically, we have added detail pertaining to the enterprise budgets, particularly the section related to cost estimates, elaborated on the net present value analysis and risk model, and expanded the discussion section. With respect to the net present value analysis, table 3 now illustrates discounted net returns (i.e., net returns after applying net present value analysis) rather than the undiscounted net returns. This change makes it easier to understand the results presented in table 5.

Lines 4-10: correct the 1-3 indication as 1 and 2.

Corrected.

Lines 218-219: a clear conclusion and extra information will provide more clarity on the method and analysis carried out.

In response to your concerns, we have rewritten and expanded the first two paragraphs in the discussion section.

Additional Reply:

In the process of addressing comments from another reviewer would have added text to the abstract and discussion sections. These additions, along with addressing your comments, have improved the manuscript.

Round 2

Reviewer 2 Report

N/A